# Efficient Hydrolysis of Earthworm Protein and the Lipid-Lowering Mechanism of Peptides in the Hydrolysate

**DOI:** 10.3390/foods14132338

**Published:** 2025-07-01

**Authors:** Mengmeng Zhang, Xiang Mai, Shanghua Yang, Yuhua Huang, Lina Zhang, Wenbin Ren, Weidong Bai, Xuan Xin, Wenhong Zhao, Lisha Hao

**Affiliations:** 1College of Chemistry and Chemical Engineering, Zhongkai University of Agriculture and Enineering, Dongsha Street 24, Guangzhou 510225, China; zhangmengmengscut@126.com; 2College of Light Industry and Food Technology, Zhongkai University of Agriculture and Engineering, Dongsha Street 24, Guangzhou 510225, China; mx56572025@163.com (X.M.); annabelyoung09@gmail.com (S.Y.); 15119309783@163.com (Y.H.); rwbzk@126.com (W.R.); whitebai2001@163.com (W.B.); 3College of Food Sciences and Engineering, South China University of Technology, Guangzhou 510640, China; linazhang1127@163.com; 4Guangdong Key Laboratory of Science and Technology of Lingnan Special Food, Zhongkai University of Agriculture and Engineering, Dongsha Street 24, Guangzhou 510225, China; 5School of Food Science and Technology, Jiangnan University, Wuxi 214122, China; foodbiolisa@163.com

**Keywords:** earthworm, peptide, autolysis, lipid-lowering activity, inflammation

## Abstract

Earthworms are valued as a dietary protein source in many regions. Earthworm protein can yield bioactive peptides, but enzymatic hydrolysis is inefficient by commercial proteases, and bioactivity development is still inadequate. This study developed a novel efficient method for degrading earthworm protein and investigated the lipid-lowering activity and mechanism of earthworm peptides. It was found that combining autolysis and alcalase exhibited a higher hydrolysis degree of earthworm protein of 43.64 ± 0.78% compared to using autolysis or alcalase only. The hydrolysate significantly reduced lipid accumulation in steatotic hepatocytes. LC-MS/MS results showed that the primary lipid-lowering peptides (EWPs) in the hydrolysate were small molecule peptides with molecular weights of 500–1000 Da and chain lengths of 4–7 amino acid residues. Western blot results demonstrated that EWP regulated the expression of lipid metabolism-related proteins, including APOC3, HMGCR, PCSK9, SREBP1, C/EBP-α, NPC1L1, PPAR-γ, and CYP7A1. Transcriptomic analysis and validation experiments indicated that the lipid-lowering activity of EWP was associated with its suppression of inflammatory factors, such as IL-6. This study presents an efficient enzymatic hydrolysis strategy for earthworm protein utilization, laying the foundation for its application in functional foods such as protein supplements, nutraceutical capsules, hypoallergenic infant formulas, and sports nutrition products.

## 1. Introduction

Terrestrial invertebrates, including *Tenebrio molitor* L. and *Bombyx mori* L., are potential protein alternatives because of their high nutritional value and eco-friendly characteristics. Earthworms are terrestrial invertebrates with rich protein (56–66%, dry weight), making them a potential source of dietary protein for humans. Earthworm protein is regarded as a novel food resource in China. In fact, earthworm-eating traditions persist across circum-Pacific regions, especially among southeastern Chinese coastal communities, Philippine archipelago inhabitants, and Australian Aboriginals. Recently, earthworm farming technology on an industrial scale has facilitated the application of earthworm as food. In addition to being used as a protein-rich raw material for food products such as hamburgers, biscuits, and canned foods, earthworm protein can also be processed into various functional bioactive peptides, such as antioxidant peptides, immunomodulatory peptides, and antihypertensive peptides [1,2,3]. However, it is difficult to obtain a high degree of hydrolysis using commercial proteases for earthworm protein [4]. The underlying causes may include the poor solubility of earthworm protein and the low substrate affinity of commercial proteases toward this substrate. Developing highly efficient enzymatic hydrolysis methods for earthworm protein holds significant implications for the preparation and application of novel earthworm-derived bioactive peptides.

Autolysis constitutes an evolutionarily conserved self-digestion mechanism across eukaryotic and prokaryotic organisms. Upon structural disruption of the organism leading to the release of internal components, endogenous enzymes break down their proteins. This autolytic phenomenon is observed in earthworm, fish, shrimp, squid, and crab; furthermore, studies have been conducted to explore the utilization of autolysis for preparing bioactive peptides [5]. Endogenous enzymes demonstrate superior substrate affinity toward native proteins. However, using autolysis to prepare active peptides requires a longer time because of the low abundance of endogenous proteases. Prolonged autolysis may elevate the risk of microbial contamination in enzymatic hydrolysates. In this study, it is speculated that the combination of autolysis and commercial proteases may compensate for their respective deficiencies.

Lipids play a crucial role in human physiology. Nonetheless, variations in lipid composition have profound impacts on cellular functions, the immune system, antioxidant defenses, and inflammatory responses [6]. Abnormalities in lipid metabolism can trigger dyslipidemia, inflammation, oxidative stress, insulin resistance, and abnormal cellular activation. These conditions progressively manifest as metabolic syndromes, including hyperlipidemia, diabetes, type 2 diabetes, and nonalcoholic steatohepatitis, or even induce neurodegenerative diseases and cancer in more severe cases [7]. Pharmacological lipid-lowering remains the primary therapeutic approach in modern dyslipidemia management frameworks. Clinically, statins, fibrates, and novel PCSK9 inhibitor drugs are widely used to regulate blood lipids. These medications effectively reduce plasma cholesterol levels but are accompanied by elevated creatine kinase levels, myalgia, and gastrointestinal reactions. Developing safe and effective natural lipid-lowering components has thus garnered considerable research interest. Research has reported that earthworm-rich feed and derived extracts regulate lipid metabolism in rats and guinea pigs [8,9]. However, no studies have been reported on the development of lipid-lowering peptides derived from earthworm protein.

Herein, we conducted a two-step enzymatic hydrolysis of earthworm proteins involving autolysis followed by treatment with alcalase. The study further assessed the lipid-lowering activity of the earthworm protein hydrolysates using a steatosis hepatocyte model. The amino acid sequences of the lipid-lowering peptides in the enzymatic hydrolysates were subsequently identified. Further mechanistic analysis of hypolipidemic peptides was conducted using transcriptomics and Western blot. This study establishes an innovative framework for preparing active peptides from earthworm proteins. It further reveals the potential mechanism of action of earthworm lipid-lowering peptides, which lays a foundation for utilizing earthworm resources in functional foods.

## 2. Materials and Methods

### 2.1. Materials and Chemicals

Mature Eisenia fetida earthworms (10–12 weeks old, cattle manure-fed) were obtained from Dinglong Biotech (Guangzhou, China). Alcalase and Interleukin-6 (IL-6), and CCK-8 kits were obtained from Yuanye Biotech (Shanghai, China) and MedChemExpress (Shanghai, China), respectively. Sephadex G-15, simvastatin, and lipid profile assay kits (triglycerides, total cholesterol, LDL-C, HDL-C) were supplied by Solarbio (Beijing, China). Cell proliferation/toxicity and modified Oil Red O staining kits were sourced from Melone Pharma (Dalian, China) and Beyotime (Shanghai, China). Hepatic function kits (ALT, AST) were purchased from Nanjing Jiancheng Bioengineering (Nanjing, China). The antibodies purchased included β-actin (Abcam, Shanghai, China); HMGCR, PCSK9, PPAR-γ, SREBP1, and C/EBPα (ABclonal, Wuhan, China); APOC3, NPC1L1, and CYP7A1 (Proteintech, Wuhan, China). HPLC-grade solvents were used for chromatography, while the other chemicals used were of analytical grade.

### 2.2. Earthworm Protein Hydrolysate Production (EPH)

Eisenia fetida were homogenized with ultrapure water at a ratio of 1:2 (*w*/*v*). The homogenate was stirred for 2 h using a magnetic stirrer at 40 °C and was then heated for 10 min at 100 °C in a water bath to stop autolysis. The pH of the homogenate was adjusted to 9.5, followed by the addition of alcalase (6.0 × 10^4^ U) and incubation for 2 h at 50 °C. The mixture was then heated at 100 °C for 10 min to terminate enzymatic hydrolysis and then subjected to refrigerated centrifugation (10,000× *g*, 10 min). The autolyzed product was used after 4 h, while the enzymatic hydrolysate obtained through alcalase hydrolysis for 4 h was employed as a control. The supernatant was collected, dialyzed through a dialysis bag (cutoff, 100 Da), and freeze-dried.

### 2.3. Degree of Hydrolysis (DH)

The DH was determined using a modified method [1]. The earthworm protein was first subjected to acid hydrolysis by mixing 10 mg of the protein with 4.0 mL of HCl solution (6 mol/L), followed by sealing in an ampoule bottle and hydrolyzing at 110 °C for 24 h to obtain complete hydrolysate. The two-step hydrolysate and the acid hydrolysate (1 mL each) were separately mixed with equal volumes of phosphate-buffered saline (PBS, pH 8.2) and 0.05% (*w*/*v*) 2,4,6-trinitrobenzenesulfonic acid (TNBS), followed by incubation at 50 °C for 1 h in the dark. The reaction was quenched with 2 mL 0.1 M HCl and diluted with 5 mL deionized water. The absorbance of the reaction mixture was measured at 340 nm using a 752S spectrophotometer (Lengguang, China). Calibration curves were subsequently generated using L-leucine standards. DH was calculated using Equation (1).DH = [(Ht − H0)/(Hmax − H0)] × 100%(1)
where Ht = α-amino acid nitrogen content at time t (mmol/L), H0 = initial α-amino nitrogen content (mmol/L), and Hmax = total amino acids after 24 h acid hydrolysis (6 M HCl, 110 °C).

### 2.4. Soluble Peptide Yield (YSP)

The soluble peptide yield was determined using an optimized experimental protocol [1]. Briefly, samples and BSA standards (0–250 μg/mL) were mixed with 10% (*w*/*v*) trichloroacetic acid (1:1, *v*/*v*) and incubated at 25 °C for 10 min. The mixtures were centrifuged (8000× *g*, 10 min), and the supernatants were sequentially mixed with Folin–Ciocalteu A reagent (30 °C, 30 min; 1:5 *v*/*v*) and Folin–Ciocalteu B reagent (25 °C, 30 min; supernatant: A: B = 1:5:2 *v*/*v*/*v*). Absorbance measurements used to establish standard curves using glutathione (GSH) (0–2 mg/mL) and bovine serum albumin (BSA) were taken at 340 nm and 500 nm, respectively. These curves were established to quantify soluble peptide content (Q1) and total protein concentration (Q2). YSP was calculated using Equation (2).YSP = (Q1/Q2) × 100%(2)
where Q1 and Q2 represent soluble peptide (μg/mL) and total protein (μg/mL) concentrations in the sample.

### 2.5. Molecular Weight Distribution

Samples were co-analyzed using molecular weight calibration standards: cytochrome C (12,384 Da), peptidase (6500 Da), baculopeptide (1422 Da), Gly-Gly-Tyr-Arg (451 Da), and Gly-Gly-Gly (189 Da) after ultrasonication in the mobile phase (5 min) and 0.45 μm membrane filtration. Size exclusion chromatography was performed using a Waters HPLC system equipped with a TSK-gel 2000 SWXL column (300 × 7.8 mm) under optimized chromatographic conditions: acetonitrile/water (40:60, *v*/*v*) containing 0.1% trifluoroacetic acid as mobile phase: UV 220 nm, flow: 0.5 mL/min, temp: 30 °C, and injection: 20 μL.

### 2.6. Isolation of EPH

The EPH sample was dissolved to a concentration of 1 mg/mL and subsequently pre-filtered through a 0.45 μm membrane. The samples were then sequentially fractionated using 10 kDa and 3 kDa molecular weight cutoff (MWCO) ultrafiltration membranes, yielding three fractions (>10 kDa, 3–10 kDa, and <3 kDa) that were lyophilized for preservation. The <3 kDa lyophilized fraction was reconstituted in distilled water, filtered through a 0.45 μm membrane, and chromatographed through a Sephadex G-15 column using 2 mL of 5 mg/mL sample solution. The components were eluted with distilled water at a flow rate of 0.5 mL/min. Samples were automatically collected every 5 min, and the peptide content was subsequently quantified by measuring the absorbance at 220 nm. Target fractions were lyophilized for preservation.

### 2.7. L02 Cell Culture, Modeling, and Treatment

L02 hepatocytes were maintained in DMEM containing 10% fetal bovine serum and 1% penicillin–streptomycin at 37 °C and 5% CO_2_. Cells were seeded into 96- or 6-well plates (2 × 10^4^ cells/well) and subjected to serum deprivation for 24 h. The cells were subsequently inducted with a 1 mM free fatty acid (oleic acid: palmitic acid = 2:1 mol/mol) for 24 h to establish a lipid accumulation model. The high-fat model was supplemented with experimental samples (0.5 mg/mL) for 24 h, with 100 μM simvastatin serving as the standard for comparison. Cellular viability was assessed using CCK-8 through spectrophotometric measurement at 450 nm and was expressed as the percentage of the untreated controls. Triglyceride (TG), total cholesterol (TC), lipoprotein fractions (HDL-C/LDL-C), and hepatic enzymes (ALT/AST) were quantified using commercial assay kits.

### 2.8. Oil Red O Staining and Quantitative Analysis

L02 cells were subjected to PBS washing cycles, fixed with chilled 10% formalin (20 min), and stained with 0.5% Oil Red O solution (isopropanol: water = 3:2, *v*/*v*) for 20 min. Residual dye was removed by washing with 60% isopropanol for 2 min, followed by rinsing thrice with distilled water. Cells were counterstained with hematoxylin for microscopic analysis. Oil Red O retained in stained cells was solubilized with 0.5 mL DMSO for quantitative lipid analysis. The absorbance of the cells was measured at 490 nm using a BioTek microplate reader. Data normalization was performed using the following formula:

Normalized ORO intensity = (OD490s/OD490m) × (CVm/CVs), where s and m denote sample-treated and untreated model control groups, respectively. OD490 represents absorbance at 490 nm, while CV corresponds to cell viability measured using the CCK-8 assay.

### 2.9. Sequence Identification Analysis by LC-MS/MS

Bioactive peptide fractions isolated through Sephadex G-15 gel filtration were subjected to structural characterization using liquid chromatography–tandem mass spectrometry (LC-MS/MS). Raw spectra were processed using the PEAKS Studio 10.6 De novo platform. The search parameters were as follows: fixed modification of carbamidomethylation (C), variable modifications including methionine oxidation (M) and N-terminal acetylation, unspecific enzymatic cleavage with up to 2 missed cleavage sites, mass error tolerances set at 20 ppm for precursor ions, and 0.02 Da for fragment ions.

The purified earthworm crude peptide samples were subjected to reductive alkylation using 0.1% FA. The conditions of nanoLC (Easy-nLC 1200) for LC-MS/MS were as follows: Acclaim PepMap RPLC C18 (precolumn: 300 μm × 5 mm, 5 μm; analytical column: 150 μm × 150 mm, 1.9 μm); mobile phase A: 0.1% formic acid aqueous solution; mobile phase B: 80% ACN/0.1% formic acid aqueous solution; flow rate: 600 nL/min; gradient elution program: 0~2 min, B: 4%~8%; 2~45 min, B: 8~28%; 45~55 min, B: 28~40%; 55~56 min, B: 40~95%; 56~66 min, B: 95%. The Q Exactive™ Hybrid Quadrupole-Orbitrap™ Mass Spectrometer (Obitrap Fusion Lumos) (Thermo Fisher Scientific, Waltham, MA, USA) was used. It was set at a scanning range of m/z 100~1500, Nanoflow NSI Positive Ion Source, Ionization Voltage of 2600 V, and temperature of 350 °C.

### 2.10. Western Blotting

Cells were lysed with NP40 buffer and centrifuged (12,000× *g*, 10 min, 4 °C) to harvest supernatants. The protein extracts were separated using SDS-PAGE and subsequently transferred onto nitrocellulose membranes. The membranes were then blocked with 5% BSA for 2 h at ambient temperature. Sequential incubations included primary antibody incubation at 4 °C overnight and HRP-conjugated goat anti-rabbit secondary antibody incubation at 25 °C for 1 h post-TBST washing. The protein bands were visualized using enhanced ECL chemiluminescence after three TBST wash cycles.

### 2.11. Transcriptome Analysis

Analysis of the earthworm transcriptome followed RNA-Seq library construction and on-board sequencing. Two cDNA strands were synthesized by reverse transcription using mRNA-enriched short fragments as templates. The cDNA strands were then subjected to end repair, poly (A) addition, ligation junction treatment, and selective PCR amplification to obtain the libraries, which were subsequently quantified and quality controlled. The libraries were then sequenced using a gene sequencer, and the results were stored in the fastq format to generate raw reads.

Data quality control and sequence comparison were performed by filtering the raw data using fastp software (V0.21.0) to remove splices and unqualified sequences to obtain clean reads. The clean reads were quality controlled (QC) using FastQC software and compared to the earthworm reference genome to identify and analyze the genes. The earthworm reference genome sequences and gene model annotation files were downloaded from the NCBI database and HISAT2 (V2.1.0) and were used to construct the reference genome index. The quality control results are presented in the Appendix A.

The relative quantitative and enrichment analysis of genes was performed through comparative quantification of inter-group gene expression differences, followed by the measurement of transcript abundance using Read Count values derived from String Tie software (v2.1.5). FPKM (Fragments Per Kilobase of transcript per Million mapped reads) was employed as a normalization metric. This parameter standardized gene-specific fragment counts through dual calibration against both transcript length and total sequencing depth, achieving dimensional uniformity in quantification of the expression level. Functional enrichment analysis of differentially expressed genes (DEGs) was performed using DESeq2 (v1.30.1), followed by inter-group comparisons between HFT and EWP groups. DEGs were identified based on FPKM expression profiles. The selection criteria were a |fold change| ≥ 2 and *p* < 0.05. The transcriptional regulation direction was classified based on log2FC values: upregulated genes were those with log2FC > 1 and *p* < 0.05, while the downregulated genes were those with log2FC < −1 and *p* < 0.05. GO term enrichment analysis was conducted using the hypergeometric distribution algorithm in the clusterProfiler package (v3.8.1) to identify significantly enriched biological functions (*p* < 0.05). KEGG pathway analysis specifically targeted the statistically significant pathways (*p* < 0.05).

The interactions between proteins were analyzed using the String (V11.5) database (https://string-db.org/ (accessed on 7 June 2024) and the R package STRINGdb (V2.8.4) software. The top 100 proteins were selected to draw a network diagram.

### 2.12. Real-Time Quantitative Polymerase Chain Reaction Validation (RT-qPCR)

Five target genes were analyzed using SYBR Green-based qPCR to validate RNA-seq data reliability. Total RNA isolation was performed using Trizol reagent (Invitrogen, Carlsbad, CA, USA)), followed by cDNA synthesis with HiScript^®^ II Reverse Transcriptase (Vazyme Biotech Co., Ltd., Nanjing, China). The primer sequences used were designed using Primer Premier 5.0 and are outlined in Appendix A. Amplifications were conducted on a TL-988 Real-Time PCR system (Xi’an Tianlong Science and Technology Co., Ltd. (Xi’an, China)). The amplification conditions were an initial denaturation stem at 95 °C for 90 s; followed by 40 cycles of denaturation, primer annealing, and extension at 95 °C for 5 s; 60 °C for 15 s; and 72 °C for 20 s, respectively. The PCR reaction mixtures contained 10 μL of 2 × ChamQ SYBR Master Mix, 0.4 of μL forward/reverse primers (10 μM each), 2 μL of cDNA template, and 5.2 of μL nuclease-free H_2_O, totaling 20 μL. The GAPDH served as an endogenous control. The relative quantification of gene expression data was performed using the 2^−ΔΔCT^ method.

### 2.13. Statistical Analysis

Statistical analysis was performed using One-way ANOVA and Duncan’s multiple-range test by the SPSS Statistic 26.0 software (SPSS, Inc., Chicago, IL, USA). *p* < 0.05 was considered statistically significant. All data are presented as means of three replicates ± standard deviation (SD). The experimental results were plotted using Prism 8.0 software.

## 3. Results

### 3.1. Hydrolysis of Earthworm Proteins

The degree of hydrolysis (DH) quantifies the cleavage efficiency of polypeptide bonds during proteolysis. Protein hydrolysis by proteases produces free amino acids and peptides [10]. The degree of protein hydrolysis positively correlates with the production of small-molecular-weight peptides during the reaction. The soluble peptide yield (YSP) can thus be used to measure the degree of protein hydrolysis. Herein, the DH and YSP (Figure 1), specifically from the autolysis process alone, of the earthworm protein product were 25.98 ± 0.13% and 66.78 ± 1.57%, respectively. Zhang et al. reported a 22.38% hydrolysis degree and 77.92% acid-soluble peptide yield of earthworm digests, using autolysis [2]. This result is consistent with the results obtained herein. The DH and YSP obtained after enzymatic hydrolysis with alcalase were 20.38 ± 0.54% and 77.63 ± 0.91%, respectively. Similarly, Zhao et al. achieved a hydrolysis degree of less than 12% when using alcalase, papain, trypsin, neutrase, and protame to enzymatically hydrolyze Lumbricus protein [4]. Alcalase, belonging to the serine protease family, efficiently hydrolyzes protein peptide bonds in alkaline environments (pH 9–11). Its cleavage sites primarily target the carboxyl terminal of hydrophobic or aromatic amino acid residues within protein peptide chains. Earthworm protein is rich in collagen, whose tightly wound triple-helical structure creates steric hindrance. This causes hydrophobic or aromatic amino acid residues to be buried within the protein interior, obstructing alkaline protease’s access to cleavage sites. Consequently, alkaline protease exhibits a low degree of hydrolysis toward earthworm protein. In this study, the DH and YSP, achieved using the two-step enzymatic hydrolysis method, were 43.64 ± 0.78% and 82.88 ± 0.78%, respectively. These results were significantly higher than those obtained using autolysis and alcalase only. Studies indicate that pepsin and trypsin facilitate the digestion of earthworm proteins, achieving a hydrolysis degree of 22.91% and a soluble peptide yield of 79.19% [1]. Lin et al. reported a 13.13% hydrolysis degree and 88.66% soluble peptide yield using enzymatic hydrolysis with pepsin and trypsin [3]. The results demonstrate that the two-step enzymatic method used in this study is more efficient compared to methods that solely utilize autolysis or exogenous protease. This efficiency is attributed to the intrinsic proteases of earthworms, which exhibit a high affinity for earthworm proteins, causing them to degrade easily. However, the degree of enzymatic degradation is limited by the finite availability of endogenous proteases within the earthworm. Commercially available proteases can be used at higher doses but are less efficient at proteolytic digestion because of the solubility and dispersion of earthworm proteins in the enzymatic hydrolysis buffer system. Autolysis serves to preliminarily degrade earthworm proteins, thereby enhancing their solubility and dispersibility and facilitating the action of commercial proteases on these proteins when the two methods are combined. The two-step enzymatic digestion employed in this study exhibits potential for industrial application because of its simplicity and cost-effectiveness.

The HPLC-SEC method was used to examine the molecular weight distribution in the earthworm protein hydrolysates to evaluate the enzymatic hydrolysis effect of the two-step enzymatic digestion method. The volume-exclusion high-performance liquid chromatography of EPH is shown Appendix A. The contents of the >10 kDa, >5 kDa, <3 kDa and <180 Da fractions of earthworm proteolytic digests obtained from the two-step enzymatic preparation were 0.91%, 3.91%, 94.2% and 25.88%, respectively (Table 1). Small-molecule peptides typically contain 2~20 amino acid residues and have a molecular weight of less than 3000 Da [11]. Free amino acids typically have molecular weights below 180 Da. Notably, the primary constituents of EPH are small molecular peptides and free amino acids, with small molecular peptides (180–3000 Da) accounting for 68.32% of the total. Small molecular peptides usually exhibit functional properties, such as antioxidant, ACE inhibitory, immunomodulatory, and lipid-lowering activity, highlighting the good biological activity of EPH [1,3,12].

### 3.2. Lipid-Lowering Activity of EPH and Its Ultrafiltration Components

The L02 hepatocyte-based steatosis model is commonly used to evaluate lipid-lowering activity. Herein, ultrafiltration components of different molecular weights did not affect the viability of L02 cells at 500 μg/mL (Figure 2A). The effects of EPH and its various ultrafiltration fractions on hepatocyte lipid accumulation were evaluated. Treatment with free fatty acid led to the accumulation of a large amount of fat droplets in cells of the HFA group. In contrast, cells in the negative control group did not accumulate lipid droplets (Figure 2B). This finding demonstrates the successful establishment of the hepatocyte steatosis model. Notably, the accumulation of intracellular lipid droplets was significantly reduced after treatment with EPH and its ultrafiltration components compared to the HFA group, suggesting that EPH and its ultrafiltration components possess good lipid-lowering activity.

The accumulation of intracellular lipid droplets was quantitatively analyzed to accurately assess the lipid-lowering effects of EPH and its ultrafiltration components (Figure 2C). The amount of lipid droplet accumulation in the EPH and its ultrafiltration component groups (3–10 kDa, <3 kDa) was significantly lower than that in the HFA group, which is consistent with the results shown in Figure 2B. The <3 kDa fraction showed the best lipid-lowering effect among the ultrafiltration components. Studies demonstrate that <3 kDa fraction from other protein sources exhibit better lipid-lowering effects. For example, Wei et al. demonstrated that <3 kDa of corn peptides effectively decreased lipid accumulation and inhibited the accumulation of reactive oxygen species (ROS) and oxidative stress in L02 cells, thereby providing potential preventive and adjunctive therapeutic value for nonalcoholic fatty liver disease [13]. Ye et al. reported that effective lipid-lowering peptides derived from tea proteins were also highly concentrated in the <3 kDa fraction. This result is attributed to the <3 kDa fraction primarily comprising small molecule peptide [14]. Of note, <3 kDa fractions exhibited a statistically significant advantage in hypolipidemic efficacy at 100 μM simvastatin. Simvastatin is a lipid-lowering drug widely used in clinical practice. However, the long-term use of simvastatin has adverse effects, such as liver function damage, muscle damage, and digestive issues. Earthworm proteins have been historically utilized as dietary components across multicultural food systems. The <3 kDa fraction is thus superior to simvastatin in terms of safety. However, the main components of EPH and its ultrafiltration components are peptides, which can be degraded by pepsin and trypsin during gastrointestinal digestion. The potential of EPH and its ultrafiltration components to induce lipid-lowering activity after digestion thus needs further investigation.

Free fatty acids are mainly converted into neutral fats, such as triglycerides (TG) and cholesterol (TC), after digestion and absorption. Cholesterol is bound to lipoproteins and transported in the bloodstream during metabolic processes. High-density lipoproteins (HDL) are primarily produced in the liver. They transport cholesterol from other tissues outside the liver for metabolism, leading to a negative correlation between high-density lipoprotein cholesterol (HDL-C) levels and lipid metabolic disorders. Low-density lipoproteins (LDL) are the primary carriers of endogenous cholesterol. The excessive accumulation of LDL cholesterol (LDL-C) in the bloodstream can clog blood vessels, leading to atherosclerosis. TG, TC, LDL-C, and HDL-C were examined to evaluate the lipid-lowering effect of EPH and its ultrafiltration fractions. TG, TC, and LDL-C levels in the HFA group were significantly higher, while HDL-C levels were significantly lower compared to the CK group, indicating that the cells were in a state of lipid metabolism disorder (Figure 3). EPH and its ultrafiltration components (500 μg/mL) reduced the TG, TC, and LDL-C levels, and increased the HDL-C levels, effectively alleviating the abnormalities of fatty acid-induced lipid metabolism by inducing a lipid-lowering effect. Notably, the lipid-lowering effect of EPH was superior to that of the >10 kDa and 3~10 kDa fractions. The <3 kDa fraction had the optimal effect in lowering TG, TC, LDL-C content and elevating HDL-C content.

The levels and activities of alanine aminotransferase (ALT) and aspartate aminotransferase (AST) are closely associated with hepatocellular health and are commonly clinically used to test liver function. ALT and AST increases may indicate the presence of diseases primarily characterized by liver damage. Herein, the L02 cell supernatant under the induction of FFA suggested signs of liver injury because of the significantly high levels of ALT and AST [15]. However, the levels of ALT and AST decreased after treatment with simvastatin, EPH, and its ultrafiltration components (Figure 3), restoring the normal functioning of hepatocytes.

These findings collectively suggest that the enzymatic hydrolysates of earthworm protein can alleviate fatty acid-induced lipid metabolism disorder in hepatocytes and the hepatocyte damage that arises from such disorders. Notably, the <3 kDa peptides are primarily responsible for exhibiting lipid-lowering activity.

### 3.3. Isolation and Identification of Lipid-Lowering Peptides in <3 kDa Fractions

The <3 kDa fraction containing small peptides and free amino acids was subjected to Sephadex G-15 to characterize the bioactive constituents of earthworm peptides. Of note, two fractions, E1 and E2, were obtained (Figure 4A). Figure 4B,C demonstrates that the lipid-lowering activity of the E1 fraction was significantly stronger than that of the E2 fraction. The E1 fractions were thus collected, designated as EWP, and subjected to further analysis.

The chemical structure information of bioactive peptides is closely associated with their specific bioactivities. The structure of peptides is predominantly determined by their amino acid sequences. A total ion flow chromatogram of EWP is shown in Appendix A. A total of 1359 peptides were identified in EWP, and their sequences are shown in the Appendix A. The MW of these peptides were predominantly distributed between 500–1000 Da (72%) and 1000–2000 Da (21%) (Figure 5A,B). Their lengths primarily spanned between 4 and 7 amino acids. Notably, dipeptides and tripeptides were rarely detected in the EWP fraction. The molecular weight of the peptides in the E2 fraction was smaller than that of the E1 fraction because the E2 fraction peaked later than E1. As such, dipeptides and tripeptides were potentially present in the E2 fraction. Moreover, the E2 potentially contained some free amino acids, resulting in a relatively low proportion of dipeptides and tripeptides within E2. This phenomenon was potentially the reason for the better activity of the E1 fraction than E2. An analysis of the amino acid composition at the N- and C-termini of 1359 polypeptide sequences revealed that hydrophobic amino acids accounted for 42.02% and 39.07% at the first and second positions of the N-terminus, respectively. In contrast, hydrophobic amino acids accounted for 54.75% and 54.16% at the corresponding C-terminal positions (Figure 5C). Notably, leucine (Leu), proline (Pro), and valine (Val) were the most abundant hydrophobic residues at both termini. Previous studies postulate that the hydrophobic N- and C-termini play an essential role in the lipid-lowering activity of the peptides [16]. These reports potentially explain why the E1 component exhibited strong lipid-lowering activity.

### 3.4. Effects of Earthworm Lipid-Lowering Peptides on the Levels of Crucial Proteins in Lipid Metabolism

Mechanistic investigation of the E1 fraction’s modulatory effects on hepatic lipid homeostasis involved the systematic analysis of eight critical lipid regulators: Apolipoprotein C-III (APOC3), HMG-CoA reductase (HMGCR), Proprotein Convertase Subtilisin/Kexin type 9 (PCSK9), Peroxisome Proliferator-Activated Receptor Gamma (PPAR-γ), Sterol Regulatory Element-Binding Protein 1 (SREBP1), CCAAT/Enhancer-Binding Protein Alpha (CEBPα), Niemann-Pick C1-Like 1 (NPC1L1), and Cytochrome P450 Family 7 Subfamily A Member 1 (CYP7A1) (Figure 6). APOC3 primarily functions as a potent inhibitor of lipoprotein lipase (LPL) activity, thereby delaying the breakdown of triglycerides. A reduction in APOC3 production decreases the rate of fat breakdown and contributes to low levels of triglycerides and cholesterol in the blood [17]. HMGCR is a pivotal rate-limiting enzyme in the synthesis of cholesterol in hepatocytes. Inhibition of the expression and activity of HMG-CoA reductase reduces cholesterol synthesis [18]. PCSK promotes the degradation of LDL receptors, leading to elevated levels of LDL-C in the blood [19]. SREBP regulates the synthesis of fatty acids and triglycerides by activating multiple genes involved in lipid synthesis [20]. C/EBPα functions as a multifunctional transcriptional regulator, primarily orchestrating lipid storage by activating genes involved in adipogenesis and lipid droplet formation [21]. NPC1L1 is a transporter protein essential for cholesterol uptake [22]. Treatment with fatty acids increased the expression levels of proteins involved in lipid uptake, synthesis, and accumulation, including APOC3, HMGCR, PCSK9, SREBP1, C/EBP-α, and NPC1L1. Conversely, treatment with EWP significantly reduced the levels of these proteins.

PPAR-γ primarily drives adipogenesis and lipid storage by transcriptionally activating genes involved in adipocyte differentiation and lipogenesis. Elevated PPAR-γ expression robustly promotes excessive lipid accumulation [23]. However, the activation of PPAR-γ in nonalcoholic fatty liver disease (NAFLD) mitigates hepatic lipid accumulation by enhancing fatty acid oxidation and reducing inflammatory responses. CYP7A1 serves as a rate-limiting enzyme in the classical bile acid synthesis pathway. It catalyzes the initial hydroxylation of cholesterol to 7α-hydroxycholesterol, thereby driving hepatic cholesterol catabolism and systemic cholesterol homeostasis [24,25]. Herein, fatty acid treatment downregulated the expression of proteins involved in lipid catabolism and excretion, including PPAR-γ and CYP7A1. In contrast, E1 intervention significantly upregulated their levels, highlighting its role in lipid degradation and efflux (Figure 6). Notably, lipid metabolism encompasses a multitude of proteins beyond the eight identified. Bioactive peptides exert hypolipidemic effects by targeting additional regulatory proteins involved in lipid homeostasis modulation. For instance, marine-derived collagen peptides attenuate hepatic lipid deposition in diet-induced obese mice by downregulating fatty acid synthase (FAS) and acetyl-CoA carboxylase (ACC). This phenomenon is concomitant with elevated HDL in the serum and the suppression of de novo lipogenesis [26]. These reports highlight the importance of clarifying the lipid-lowering mechanism of EWP.

### 3.5. Transcriptome Analysis

The transcriptome of steatotic hepatocytes after treatment with E1 fractions (EWP) was examined to understand the mechanism by which earthworm peptide alleviates hepatocellular steatosis. The quality control results are provided in the Appendix A.

The comparison between the control and experimental groups (HFT_VS_EWP) revealed 518 differential genes between the two groups. Among them, 214 genes were upregulated, while 304 genes were down-regulated in the EWP group compared to the HFT group. Differential genes with similar expression patterns were clustered based on their expression in different samples. Hierarchical cluster analyses were subsequently performed on the differential genes between HFT/EWP groups (Figure 7). GO functional annotation of DEGs (Figure 8A) revealed tripartite categorization encompassing biological processes (BP), molecular functions (MF), and cellular components (CC).

The enrichment analysis conducted using clusterProfiler (*p* < 0.05) revealed the top 20 significantly enriched GO terms, with a hierarchical visualization presented in Figure 8B. Only the “immune response” term met the significance threshold, suggesting that it was the most significantly affected biological process by EWP in steatotic hepatocytes.

KEGG pathway enrichment analysis of annotated DEGs identified pivotal regulatory pathways governing lipid metabolism dysregulation. Of note, 23 of the first 30 metabolic pathways screened (Figure 9A) were involved in immune responses. The top 20 most significantly enriched pathways are presented in Figure 9B. Among them, eight exhibited statistically significant enrichment. The eight included IL-17 signaling, rheumatoid arthritis, cytokine-cytokine receptor interaction, Staphylococcus aureus infection, hematopoietic cell lineage, viral protein interaction with cytokine and cytokine receptor, amoebiasis, and leishmaniasis pathways. The eight significantly enriched pathways demonstrated critical roles in orchestrating immune–inflammatory responses.

The differentially expressed genes identified in these signaling pathways (Table 2) were primarily inflammatory factors and their corresponding receptors. In both GO and KEGG analyses, relatively few genes associated with lipid metabolism were involved. This phenomenon suggests that regulating lipid metabolism-related protein levels by EWP was potentially a secondary response to alleviating inflammation. Hepatic lipid metabolism disorder and chronic inflammation form a pathological vicious cycle through bidirectional regulation. Lipid overload can trigger the release of inflammatory cytokines via multiple signaling pathways [27]. Conversely, pro-inflammatory factors inhibit fatty acid oxidation and enhance lipid synthesis, exacerbating lipid accumulation [28]. The string database analysis revealed that inflammatory factors, including *IL-6* and *CXCL8*, occupied central positions in the differential gene protein network interactions between the HFT/EWP groups (Figure 10). These cytokines exert significant regulatory effects on the expression and activity of lipid-metabolizing enzymes. Mechanistically, *IL-6* enhances *FAS* activity by disrupting insulin signal transduction [29]. *IL-6* inhibits fatty acid oxidation through a STAT3-dependent mechanism involving *PPAR-α* promoter methylation, which adjusts PPAR-α expression levels [30,31]. *IL-6* also upregulates *SREBP1* while suppressing *PCSK9* [32,33,34]. CXCL8 activates the ROS/ERK1/2 pathway to induce PPAR-γ phosphorylation at Lys268, inhibiting its transcriptional activity [35]. The expression of *APOC3*, *HMGCR*, *C/EBP-α*, *CYP7A1*, and *NPC1L1* is regulated by pro-inflammatory cytokines (TNF-α, IL-6, IL-1β, MCP-1) through NF-κB, FOXO1, STAT3, and MAPK signaling cascades [36,37]. These findings collectively suggest EWP signaling networks. The *JUN* protein (c-Jun), a core component of the activator protein-1 (AP-1) transcription complex, forms heterodimers with Fos family proteins through leucine zipper (LZ) domains to bind DNA and regulate inflammatory cytokine expression [38]. AP-1 acts as a molecular hub integrating multiple signaling pathways [39]. The receptor complex activates the Act1/TRAF6/MAPK signaling cascade upon IL-17 stimulation, leading to MAPK-mediated phosphorylation of AP-1 subunits [40]. AP-1 transcriptionally orchestrates cartilage-degrading matrix metalloproteinases (MMP1/MMP3) in the pathogenesis of rheumatoid arthritis [41]. Moreover, AP-1 directly regulates the transcription of inflammatory mediators (TNF-α, IL-2, IL-12, IL-1β, IL-8) and their receptors (IL-2R) in the cytokine–cytokine receptor interaction and Staphylococcus aureus infection pathways [42]. The transcriptomic analyses revealed that members of the Fos family, including *FosB* and *FosL1* (Table 2), exhibited significant differential expressions in the HFT and EWP groups. These findings collectively suggest that AP-1 potentially serves as a crucial transcription factor through which EWP modulates the expression of these inflammatory mediators.

### 3.6. Validation

The accuracy of the transcriptomic results was validated using qPCR, which was used to detect the expression levels of *IL-6*, *CXCL8*, *c-JUN*, *SPP1*, and *CCR1* (Figure 11). Of note, the changes in the expression levels of these genes were consistent with the transcriptomic data, confirming the reliability of transcriptome-derived gene expression profiles. Transcriptomic analysis suggested that reducing *IL-6* levels potentially played a critical role in the EWP-mediated alleviation of lipid metabolism disorders. This hypothesis was verified by co-treating steatotic hepatocytes with IL-6 and EWP. Notably, the addition of IL-6 significantly inhibited the lipid-lowering efficacy of EWP (Figure 12). Song et al. demonstrated that *Ruditapes philippinarum* bioactive peptides (RBPs) ameliorate high-fat diet-induced obesity and dyslipidemia by suppressing inflammatory responses [43]. Similarly, Liu et al. demonstrated that peptides derived from *Chlorella pyrenoidosa* exhibit both hypolipidemic and anti-inflammatory properties [44]. Our previous study revealed that earthworm protein alleviates Cyclophosphamide-induced inflammation [9]. Herein, HFT exhibited higher inflammatory cytokines than EWP. These findings collectively suggest that EWP partially mitigates hepatic lipid metabolic disturbances through anti-inflammatory mechanisms.

The mechanism by which small peptides modulate lipid metabolism and inflammatory pathways likely involves their targeted interactions with specific proteins—such as receptors, enzymes, and transporters. For instance, anti-inflammatory peptides derived from Jinhua ham bind to Toll-like receptors, restraining both the phosphorylation of key proteins in the MAPK/NF-κB pathway and the aberrant expression of inflammatory cytokines [45]. Meanwhile, bioactive peptides from Rapana venosa specifically target the TLR-NF-κB signaling axis, inhibiting the transcriptional activation of downstream inflammatory mediators [46]. The inflammatory cytokine IL-6 activates AKT2, which in turn modulates lipid metabolism through the AKT2-SREBP1-SCD1 signaling pathway [47,48]. Our previous research demonstrated that earthworm peptides bind to both TLR2 and TLR4 receptors [2]. This suggests that earthworm peptides likely ameliorate lipid metabolism dysregulation by suppressing TLR-mediated inflammatory signaling pathways. G protein-coupled receptors (GPCRs), the largest cell surface receptor superfamily, are activated by diverse ligands to couple with G proteins and trigger signaling cascades, serving as critical therapeutic targets for lipid metabolism disorders including obesity and NAFLD [49,50]. GPCR-mediated signaling regulates key transcription factors and signaling pathways such as AMPK, PPARα/γ, and SREBP-1c, thereby modulating fatty acid oxidation, synthesis, adipocyte differentiation, and lipid storage [51]. However, the specific molecular targets mediating earthworm peptides’ regulation of lipid metabolism remain to be elucidated.

It is important to note that bioactive peptides, originating from food components derived from crops, can be degraded by gastrointestinal digestive enzymes such as pepsin, trypsin, and others during digestion [52]. This degradation can modify or diminish their bioactivity. This challenge is commonly encountered with bioactive peptides prepared using commercial proteases when considering real-world applications.

In recent years, numerous studies have demonstrated that approaches like structural modification and carrier delivery systems can effectively enhance the digestive stability of peptides [53]. Therefore, a comprehensive assessment of the gastrointestinal digestive stability impacting both the structure and hypolipidemic activity of earthworm-derived peptides, coupled with the screening and development of highly effective protective technologies, constitutes an indispensable next step for their successful translation into functional food ingredients.

## 4. Conclusions

This study used combined autolysis and alcalase to efficiently hydrolyze earthworm protein. The small-molecule peptides in the earthworm protein hydrolysate alleviated lipid metabolism disorders in steatotic hepatocytes by regulating the expression levels of crucial proteins (APOC3, HMGCR, PCSK9, PPAR-Y, SREBP1, C/EBP-a, NPC1L1, CYP7A1). This mechanism potentially reduces inflammatory factors such as IL-6 through earthworm peptides. However, both the specific targets mediating the lipid-lowering effects of earthworm peptides and their gastrointestinal digestive stability require further investigation. This study provides an efficient strategy for preparing bioactive peptides from earthworm protein. It elucidates the mechanism underlying the lipid-lowering activity of earthworm peptides, laying a foundation for developing and utilizing earthworm resources.

## Figures and Tables

**Figure 1 foods-14-02338-f001:**
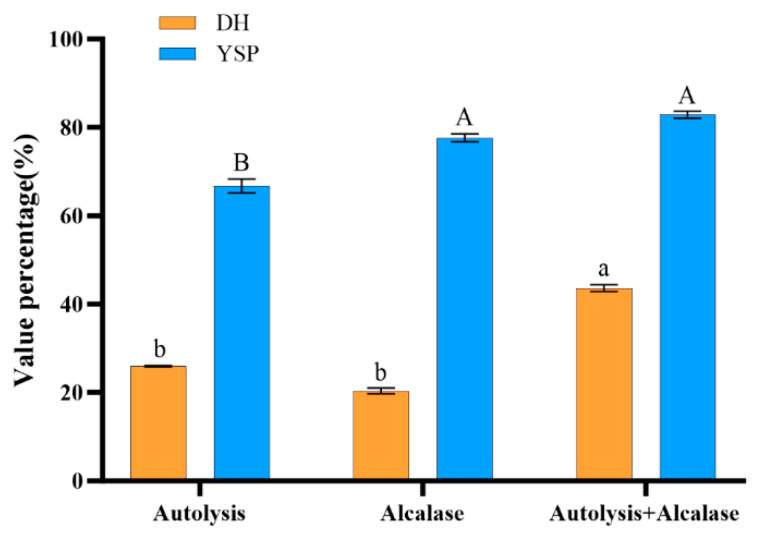
DH and YSP of different enzymatic methods. Different letters a, b and A, B indicate significant differences among groups, *p* < 0.05.

**Figure 2 foods-14-02338-f002:**
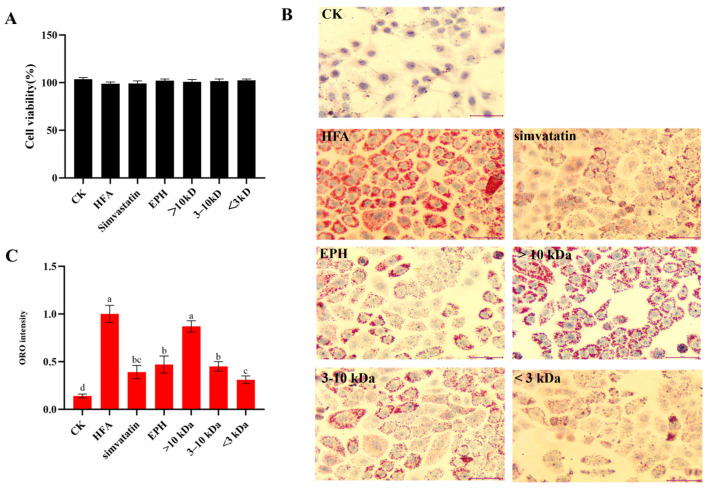
Effect of EPH and its different ultrafiltration components on the lipid accumulation in steatosis hepatocytes. (**A**) Cell viability samples of 500 μg/mL; (**B**) Images of ORO staining in cells treated with samples of 500 μg/mL; (**C**) quantitative analysis of ORO staining. CK: cells without any treatment; HFA: cells treated with high fatty acids alone; EPH: cells treated with fatty acids and earthworm protein hydrolysate; >10 kDa: cells treated with fatty acids and >10 kDa fraction in EPH; 3–10 kDa: cells treated with fatty acids and 3–10 kDa fraction in EPH; <3 kDa: cells treated with fatty acids and <3 kDa fraction in EPH. Different letters a~d indicate significant differences among groups, *p* < 0.05.

**Figure 3 foods-14-02338-f003:**
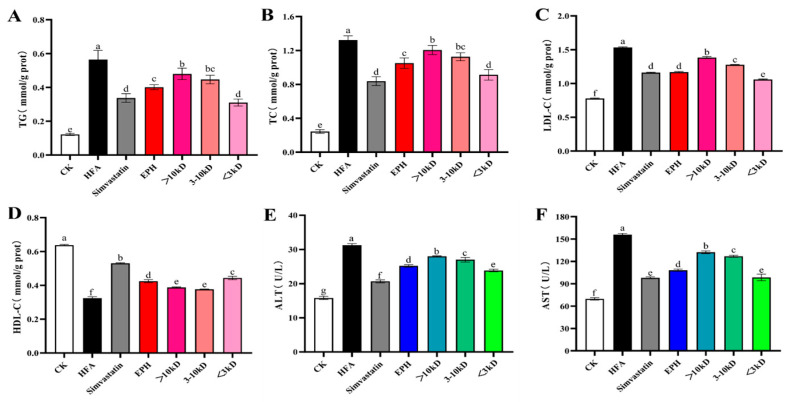
Effects of earthworm protein hydrolysate (EPH) and its different ultrafiltration components on TG (**A**), TC (**B**), LDL-C HDL-C (**C**), HDL-C, (**D**) and ALT (**E**) and AST (**F**) in steatosis hepatocytes. CK: cells without any treatment; HFA: cells treated with high fatty acids alone; EPH: cells treated with fatty acids and earthworm protein hydrolysate; >10 kDa: cells treated with fatty acids and >10 kDa fraction in EPH; 3–10 kDa: cells treated with fatty acids and 3–10 kDa fraction in EPH; <3 kDa: cells treated with fatty acids and <3 kDa fraction in EPH. Different letters a~g indicate significant differences among groups, *p* < 0.05.

**Figure 4 foods-14-02338-f004:**
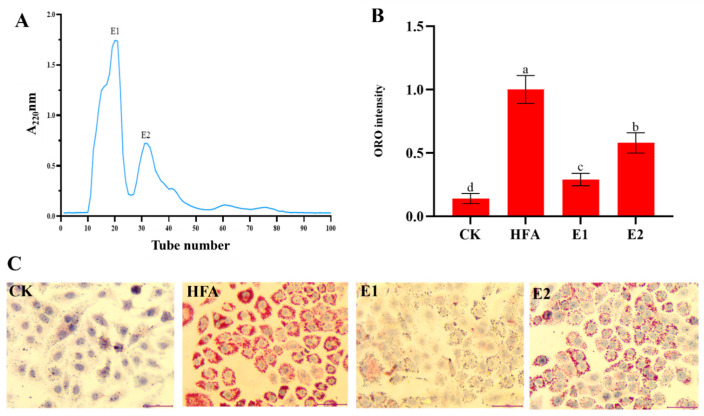
Isolation and the lipid-lowering activities of the fractions in <3 kDa components. (**A**) Gel chromatographic separation of <3 kDa components. (**B**) Quantitative analysis of ORO staining. (**C**) Images of ORO staining. CK: cells without any treatment; HFA: cells treated with high fatty acids alone; E1: cells treated with fatty acids and E1 fraction in <3 kDa components; E2: cells treated with fatty acids and E2 fraction in <3 kDa components. Different letters a~d indicate significant differences among groups, *p* < 0.05.

**Figure 5 foods-14-02338-f005:**
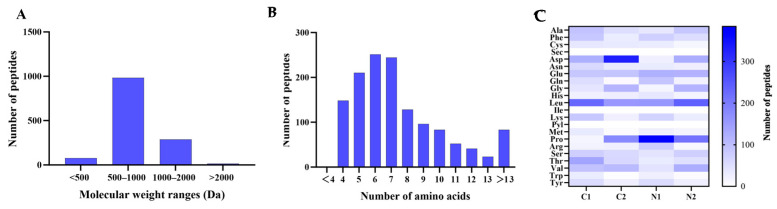
Distribution of molecular weight (**A**), amino acid number, (**B**) and terminal amino acid (**C**) of the identified peptides in EWP. C1: first amino acid from the C-terminus; C2: second amino acid from the C-terminus; N1: first amino acid from the N-terminus; N2: second amino acid from the N-terminus.

**Figure 6 foods-14-02338-f006:**
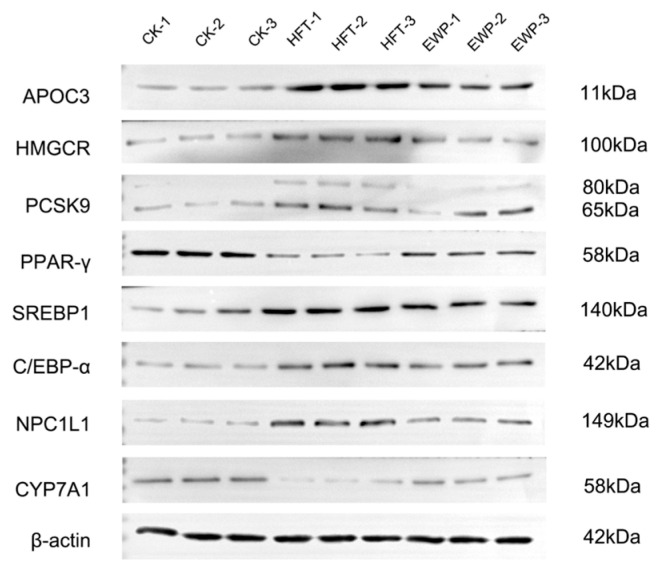
Effect of EWP on the level of proteins involving lipid metabolism.

**Figure 7 foods-14-02338-f007:**
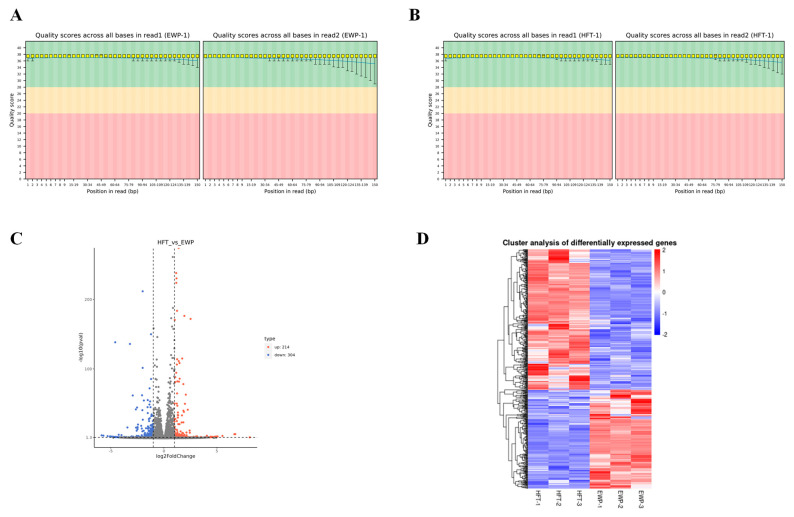
Base quality distribution (**A**) HFT group, (**B**) EWP group; Differential Gene Expression Analysis (**C**), and Cluster Dynamics between HFT and EWP treatment groups (**D**).

**Figure 8 foods-14-02338-f008:**
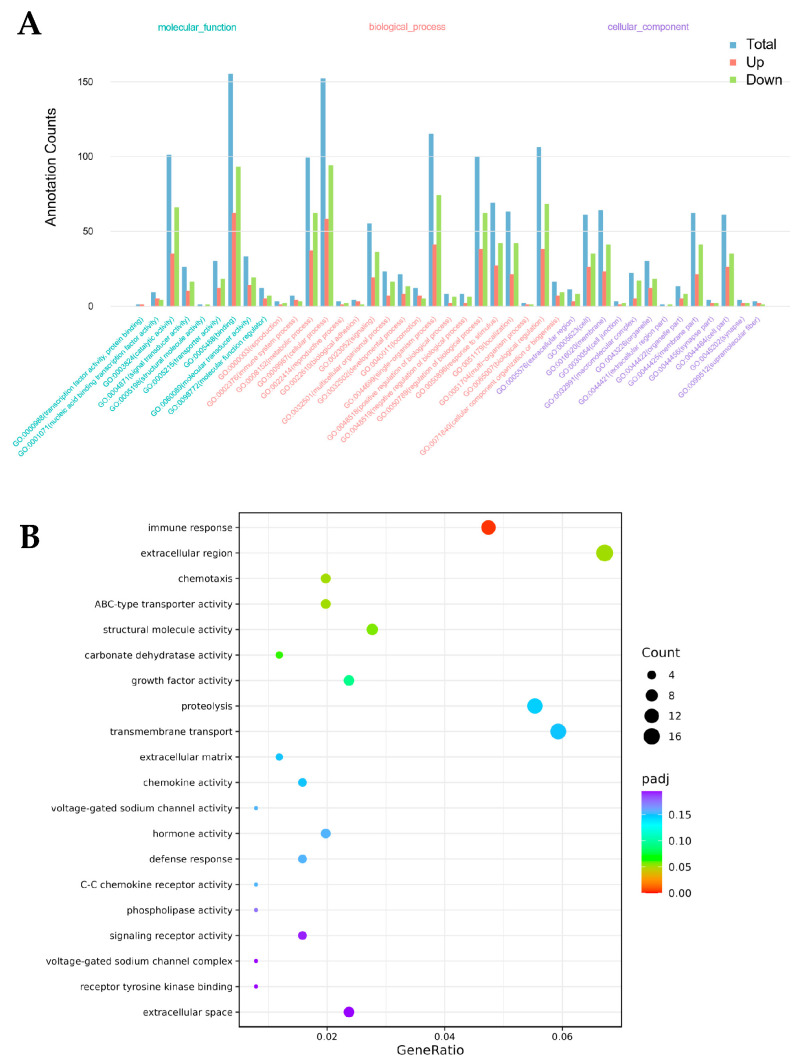
GO function annotation diagram (**A**) and GO enrichment bubble map (**B**) of differential genes of HFT/EWP group.

**Figure 9 foods-14-02338-f009:**
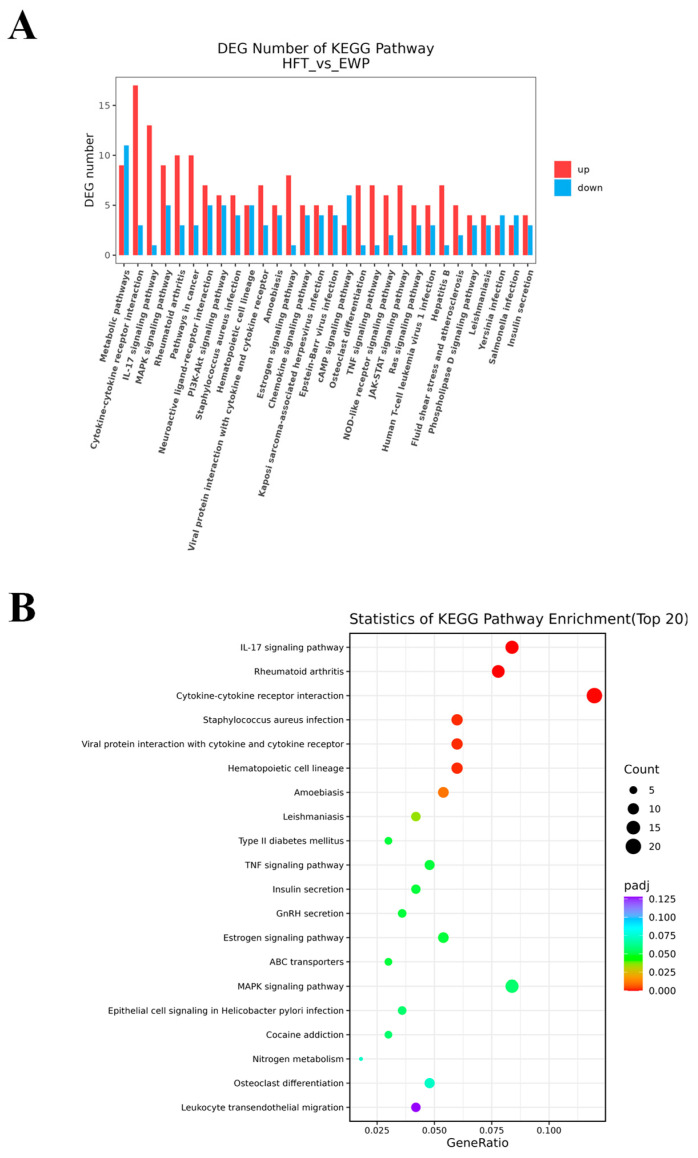
KEGG pathway annotation (**A**) and KEGG enrichment sites (**B**) of differential genes of HFT/EWP group.

**Figure 10 foods-14-02338-f010:**
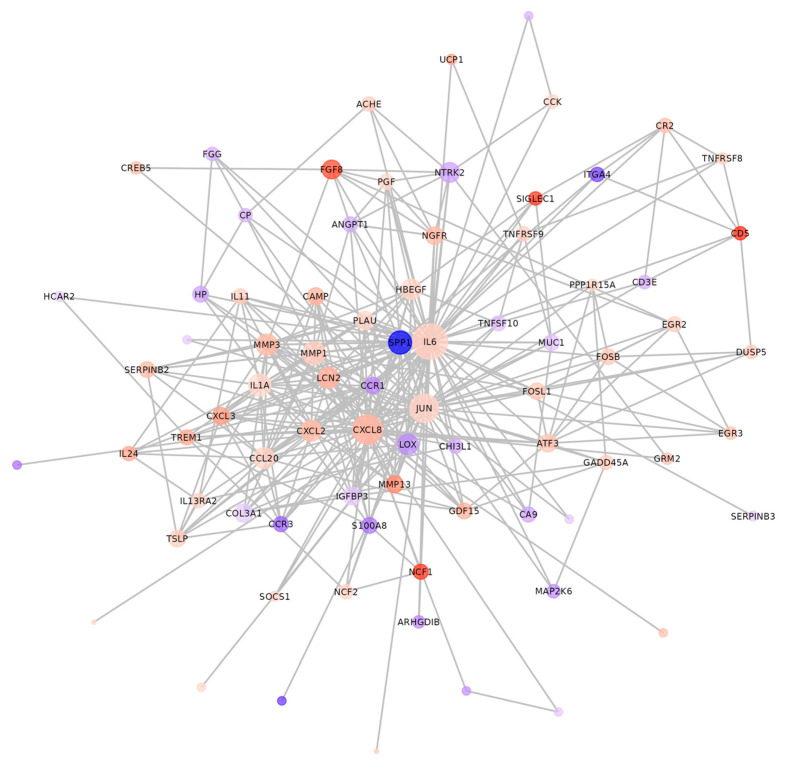
Interaction network of differential gene proteins between HFT/EWP groups. Blue = down-regulated; red = up-regulated. The color shade indicates the fold change value, with darker hues showing greater differences. The size of the dot corresponds to the number of connecting nodes, with larger dots showing higher node importance.

**Figure 11 foods-14-02338-f011:**
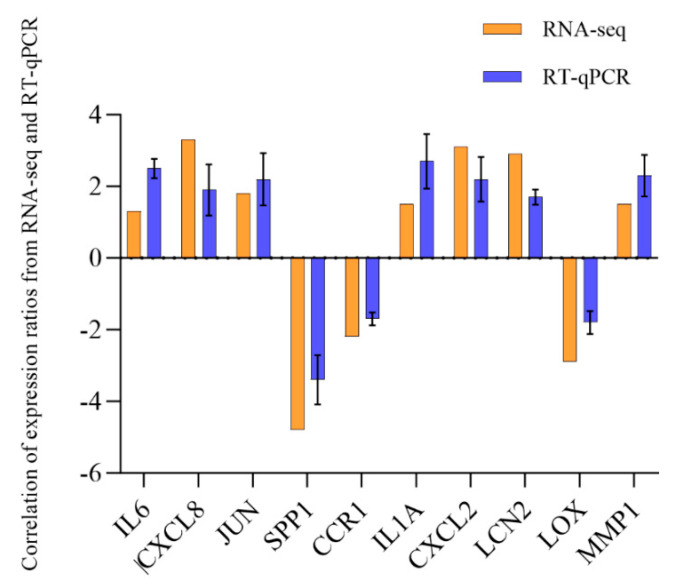
Correlation of expression ratios between RNA-seq and RT-qPCR.

**Figure 12 foods-14-02338-f012:**
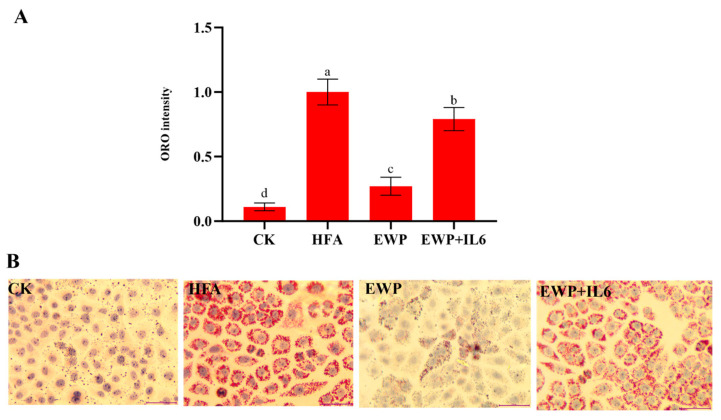
Effect of IL-6 on the lipid-lowering activity of EWP. (**A**) Quantitative analysis of ORO staining. (**B**) Images of ORO staining. CK: cells without any treatment; HFA: cells treated with high fatty acids alone; EWP: cells treated with fatty acids and earthworm peptides of E1 fraction; EWP + IL6: cells treated with fatty acids, earthworm peptides of E1 fraction and IL6. Different letters a~d indicate significant differences among groups, *p* < 0.05.

**Table 1 foods-14-02338-t001:** Molecular weight distribution of EPH.

Molecular Weight Range (Da)	Percentage of Peak Area (%)
>10,000	0.91
10,000~5000	3.00
5000~3000	1.89
3000~2000	2.66
2000~1000	8.66
1000~500	20.48
500~180	36.51
<180	25.88

**Table 2 foods-14-02338-t002:** Differentially expressed genes and associated pathways in HFT vs. EWP.

Signaling Pathway	Genes
IL-17 signaling pathway	*IL6/CXCL8/CXCL3/CXCL2/FOSL1/CCL20/JUN/LCN2/* *FOSB/MMP1/MMP3/S100A8/MAPK15/MMP13*
Cytokine–cytokine receptor interaction	*IL6/CXCL8/CXCL3/CXCL2/CCL20/IL11/TNFRSF9/GDF15/INHBA/IL13RA2/* *IL24/IL1A/NGFR/IL20/CCR3/TNFSF10/CCR1/INHBC/TSLP/TNFRSF8*
Rheumatoid arthritis	*IL6/CXCL8/CXCL3/CXCL2/CCL20/JUN/IL11/IL1A/HLA-DQB1/MMP1/MMP3/ANGPT1/HLA-DQA1*
Staphylococcus aureus infection	*KRT17/KRT16/HLA-DQB1/* *FGG/KRT14/CAMP/KRT34/LOC100653049/CFI/HLA-DQA1*
Hematopoietic cell lineage	*IL6/IL11/IL1A/HLA-DQB1/CD36/CR2/CD3E/CD5/ITGA4/HLA-DQA1*
Viral protein interaction with cytokine and cytokine receptor	*IL6/CXCL8/CXCL3/CXCL2/CCL20/IL24/IL20/CCR3/TNFSF10/CCR1*
Amoebiasis	*IL6/CXCL8/CXCL3/CXCL2/SERPINB4/SERPINB3/GNA14/PRKCG/COL3A1*
Leishmaniasis	*JUN/IL1A/HLA-DQB1/NCF2/NCF1/ITGA4/HLA-DQA1*

## Data Availability

The original contributions presented in the study are included in the article and Appendix A, further inquiries can be directed to the corresponding authors.

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
