# Peer review of "Efficient Hydrolysis of Earthworm Protein and the Lipid-Lowering Mechanism of Peptides in the Hydrolysate"

_foods, 2025, doi:10.3390/foods14132338_

Round 1

Reviewer 1 Report

Comments and Suggestions for Authors

This manuscript “Efficient Hydrolysis of Earthworm Protein and the Lipid-lowering Mechanism of Peptides in the Hydrolysate” aims to explore a novel efficient method for degrading earthworm protein and investigated the lipid-lowering activity and mechanism of earthworm peptides. However, it is important to address the following concerns through minor revisions:

Abstract: Abstract is well-structured and written. A couple of amendments will be useful for readers. Line 20-23: Split and revise the long sentence. Briefly describe the contents covered in this article, development of bioactive activities of earthworm peptide remains insufficient.

Line 36. Enlist some functional food enriched with earthworm proteins.

Line 46-48. What challenges might arise in scaling up this farming technology?

Line 65. Make up is not a suitable words, rephrase it carefully.

Results: Line 265-266. Please validate the statement with proper citation.

Line 282. Rephrase the word evidently, it’s unclear.

Line 308-310. Please validate the statement with proper citation

Reviewer 2 Report

Comments and Suggestions for Authors

Dear authors, the manuscript "Efficient Hydrolysis of Earthworm Protein and the Lipid-lowering Mechanism of Peptides in the Hydrolysate" is quite interesting and worth investigation. Please see some comments below:

1- It was very well-designed, congratulations;

2- Please double-check formatting, in particular words that should be italicized, word sources, among others;

3- Please make the enzimatic (alcalase) discussion deeper active sites, etc.

4- You have used a high resoltuion equipment (LCMS/MS), the chromatogram does not seem quite good. Have you considered other methods, maybe MALDI?

Regards

Reviewer 3 Report

Comments and Suggestions for Authors
  • In the "Materials and Methods" section. more details about the protein extraction process and consistency in raw material (like age and diet of earthworms) could enhance the understanding of the hydrolysate production.
  • Methodological Justification: Provide a rationale for selecting alcalase over other industrially available enzymes.
  • Why was 100 degree C use for inactivation of autolysis when generally 80-90 degrees is sufficient
  • Please explain in detail method used for Hmax calculation as well as any special apparatus used for the same.
  • Explain the principle behind the method used for soluble peptide yield.
  • What about the lead bioactive peptide sequences? LCMS has been mentioned. Can you provide the sequencing data of peptides (especially most bioactive).
  • Statistical Analysis Details should be included as part of the "Materials and Methods" section
  • Mechanistic Insights: Expand on the mechanistic insights into how the earthworm peptides specifically modulate lipid metabolism and inflammatory pathways thoroughly.
  • Potential consumer acceptance, regulatory hurdles, and eco-toxicity of widespread use should be assessed.
  • Referencing and Supporting Literature: Increase the number of recent references to strengthen the manuscript's current relevance, particularly in the fields of peptide stability and function.
